# Interkinetic nuclear movements promote apical expansion in pseudostratified epithelia at the expense of apicobasal elongation

**Marina A. Ferreira**[1¤], **Evangeline Despin-Guitard**[2], **Fernando Duarte**[2], **Pierre Degond**[1]\*, **Eric Theveneau**[2]\*

**1** Department of Mathematics, Imperial College London, London, United Kingdom, **2** Centre for Developmental Biology, Centre for Integrative Biology, CNRS, Université Paul Sabatier, France

¤ Current address: University of Helsinki, Department of Mathematics and Statistics, FI, Helsingin yliopisto, Finland.
\* p.degond@imperial.ac.uk (PD); eric.theveneau@univ-tlse3.fr (ET)

**Data Availability Statement:** All relevant data are within the manuscript and its Supporting Information files.

## Abstract

Pseudostratified epithelia (PSE) are a common type of columnar epithelia found in a wealth of embryonic and adult tissues such as ectodermal placodes, the trachea, the ureter, the gut and the neuroepithelium. PSE are characterized by the choreographed displacement of cells' nuclei along the apicobasal axis according to phases of their cell cycle. Such movements, called interkinetic movements (INM), have been proposed to influence tissue expansion and shape and suggested as culprit in several congenital diseases such as CAKUT (Congenital anomalies of kidney and urinary tract) and esophageal atresia. INM rely on cytoskeleton dynamics just as adhesion, contractility and mitosis do. Therefore, long term impairment of INM without affecting proliferation and adhesion is currently technically unachievable. Here we bypassed this hurdle by generating a 2D agent-based model of a proliferating PSE and compared its output to the growth of the chick neuroepithelium to assess the interplay between INM and these other important cell processes during growth of a PSE. We found that INM directly generates apical expansion and apical nuclear crowding. In addition, our data strongly suggest that apicobasal elongation of cells is not an emerging property of a proliferative PSE but rather requires a specific elongation program. We then discuss how such program might functionally link INM, tissue growth and differentiation.

## Author summary

Pseudostratified epithelia (PSE) are a common type of epithelia characterized by the choreographed displacement of cells' nuclei along the apicobasal axis during proliferation. These so-called interkinetic movements (INM) were proposed to influence tissue expansion and suggested as culprit in several congenital diseases. INM rely on cytoskeleton dynamics. Therefore, longer term impairment of INM without affecting proliferation and adhesion is currently technically unachievable. We bypassed this hurdle by generating a mathematical model of PSE and compared it to the growth of an epithelium of reference.

**Funding:** ET acknowledges support from the Fondation pour la Recherche Médicale (FRM AJE201224), the Region Midi-Pyrénées (13053025), Toulouse Cancer Santé (DynaMeca), the CNRS and Université Paul Sabatier (UMR5547). FD and EDG were supported by Toulouse Cancer Santé (DynaMeca). PD acknowledges support by the Engineering and Physical Sciences Research Council (EPSRC) under grants no. EP/M006883/1 and EP/N014529/1, by the Royal Society International Exchanges under grant no. IE160750, by the Royal Society and the Wolfson Foundation through a Royal Society Wolfson Research Merit Award no. WM130048 and by the National Science Foundation (NSF) under grant no. RNMS11-07444 (KI-Net). PD is on leave from CNRS, Institut de Mathématiques de Toulouse, France. MF acknowledges support by Imperial College, Department of Mathematics, through a Roth PhD studentship, by The Company of Biologists, Disease Models and Mechanisms, through a Travelling fellowship and by the AtMath Collaboration of the Faculty of Science of the University of Helsinki. MF short-term stays at CNRS and Université Paul Sabatier (UMR5547) were further supported by Toulouse Cancer Santé via the DynaMeca grant. The funders had no role in study design, data collection and analysis, decision to publish, or preparation of the manuscript.

**Competing interests:** The authors have declared that no competing interests exist.

Our data show that INM drive expansion of the apical domain of the epithelium and suggest that apicobasal elongation of cells is not an emerging property of a proliferative PSE but might rather requires a specific elongation program.

## Introduction

Pseudostratified epithelia (PSE) are a special type of columnar epithelia in which cells are thin and elongated. Nuclei packing is very high and forces cells to distribute their nuclei along the apicobasal axis creating multiple layers of nuclei within a monolayer of cells, hence the term pseudostratification. PSE are found across the animal kingdom from invertebrates to vertebrates [1]. During development, several structures adopt a pseudostratified configuration such as the placodes and the central nervous system in vertebrates or the imaginal discs in Drosophila. In adults, PSE can be found along the respiratory, urinary and digestive tracts (e.g. trachea, ureter, midgut) [2, 3] and various organs such as the gonads (e.g. epididymis) or the eye (lens, retina) [1]. One characteristic feature of PSE is the coordinated movements of nuclei during the cell cycle called interkinetic nuclear movements (INM) [4]. INM are decomposed in several steps: an apical to basal movement occurring during the G1 and S phases of the cell cycle and a basal to apical nuclear movement occurring during the G2 and M phases. The apical-ward movement, sometimes referred to as PRAM (Pre-mitotic Rapid Apical Movement), can be achieved via microtubules like in the chick neuroepithelium [5] or in the brain of mouse and rat embryos [6, 7] as well as in the retina of post-natal mice [8]. Such movements can also occur in an actomyosin-dependent manner as observed in the retina of fish embryos [9]. The return of nuclei to basal positions after mitoses was initially proposed to be passive and a direct consequence of nuclear crowding in the apical region of PSE. However, there are numerous evidences indicating an active role of the cytoskeleton in apical to basal nuclear displacement. For instance, Kif1A, an anterograde molecular motor of microtubules, is required for the apical to basal movement of nuclei in rat brain [7]. In addition, in mouse telencephalon, myosin II was shown to be essential for apical to basal movement [10]. Further, in ferrets' brains apical to basal movements of nuclei are faster than basal to apical movements suggesting that the nuclear movement towards basal regions of the brain is active while the opposite is observed in mouse [11]. All these observations indicate that INM are regulated by cytoskeleton-dependent mechanisms and that the actual mechanism employed differs from species to species and organ to organ.

One consequence of the cytoskeleton-based regulation of INM is that it renders INM difficult to study in vivo since it is far from the only cell process that relies on cytoskeleton dynamics. Cell-cell junctions, cell-matrix adhesions and cell contractility require normal microfilaments and microtubules dynamics. Mitosis relies on microtubules-driven separation of chromosomes and actomyosin-dependent cytokinesis. Therefore, it is currently technically impossible to study the specific roles that INM might have in PSE dynamics, growth and shape over long periods of time (hours to days) without impairing adhesions, contractility or cell division. This motivates the use of alternative approaches, such as computational modelling.

Many models of cell tissue mechanics can be found in the literature, ranging from agent-based [12] to continuum models [13, 14]. Agent-based models describe the tissue at the cell scale and have been used to study local phenomena, such as the influence of the variation of spatial constraints in the cell cycle [15], how curvature of an epithelial sheet is determined by mechanical tensions [16] or how contact inhibition of locomotion generates forces in the tissue [17]. Continuum models instead describe the system at the tissue scale (cell density) and

study global properties, such as the tissue curvature, resistance to deformation [18], contraction-elongation and tissue shear flow [13]. Despite being easier to treat, both computationally and analytically, continuum models do not incorporate all information about individual cell shape and position. An agent-based model instead is able to provide detailed spatial information and, in particular, it can account for variability in cell characteristics associated with the different stages of the cell cycle and variability in cell shape associated with the dynamics of INM (see [19] for a comparison between the continuum and agent-based frameworks). For these reasons, we opted for the agent-based approach. A large number of agent-based models of cell tissues have been developed in the last decades. The well-known Potts model [20] is a lattice-based model in which the cells may have complex shapes with a desired resolution. However, it has been reported that grid artefacts occur in cell movement and intercellular interactions [21] and they increase with particle density [22], which makes this model unsuited to describe crowded systems. Off-lattice models include for example the vertex model [15, 18, 23–25] and the Voronoi model [26, 27]. The tissue is regarded as a partition of space where each part represents one cell that is contiguous to its neighbours with no intercellular space between them. These models are able to describe densely packed systems. However, congestion is encoded into the model. In a PSE, nuclear crowding may not occur every time nor everywhere, so it should not be included in the model but rather occur as an emergent phenomenon.

Therefore, we reasoned that an appropriate framework to model a PSE would be an agent-based model where each cell moves in an off-lattice domain and interacts with its neighbours. Using such model, and comparing it with the chick neuroepithelium as a biological PSE of reference, we have explored the impact of INM, proliferation, adhesion and contractility on tissue shape, position of mitoses, pseudostratification and growth. Our results indicate that INM generate apical nuclear crowding, oppose apical shrinkage due to apical contractility and directly favor tissue growth oriented perpendicularly to the apicobasal axis (dorsoventrally, anteroposteriorly). Interestingly, all characteristics observed in the chick neuroepithelium such as apical positioning of mitoses, apical straightness, apical nuclear crowding and pseudostratification emerge from a combination of INM, proliferation, apical contractility and cell adhesion. However, the sustained linear apicobasal growth observed during development of the chick spinal cord cannot be reproduced with this combination of parameters. We show that neuroepithelial cells undergo a dramatic change of shape concomitantly to a reduction of cell volume while elongating along the apicobasal axis. This change of cell shape exceeds what is needed to accommodate new nuclei added by proliferation. Therefore, our simulations and in vivo observations strongly suggest that, while INM contribute to the expansion of the apical domain, the observed in vivo apicobasal elongation requires a specific elongation program. We then discuss whether such program, together with INM promoting apical nuclear crowding, might be a way to coordinate tissue growth and differentiation.

## Results

### Evolution of the chick trunk neuroepithelium from two to four days of development

As a biological PSE of reference, we chose the trunk neuroepithelium of the chick embryo. The neuroepithelium is a well-described PSE and has the advantage of being easily accessible for observation and manipulation. We started by monitoring the evolution of the neuroepithelium at the level of the prospective forelimb (somites 15–20) between two and four days of development [28], corresponding to developmental stage HH13- (18 somites, 48h of incubation) to stage HH23 (96 hours of incubation) and performed transversal cryosections (Fig 1A and 1B).

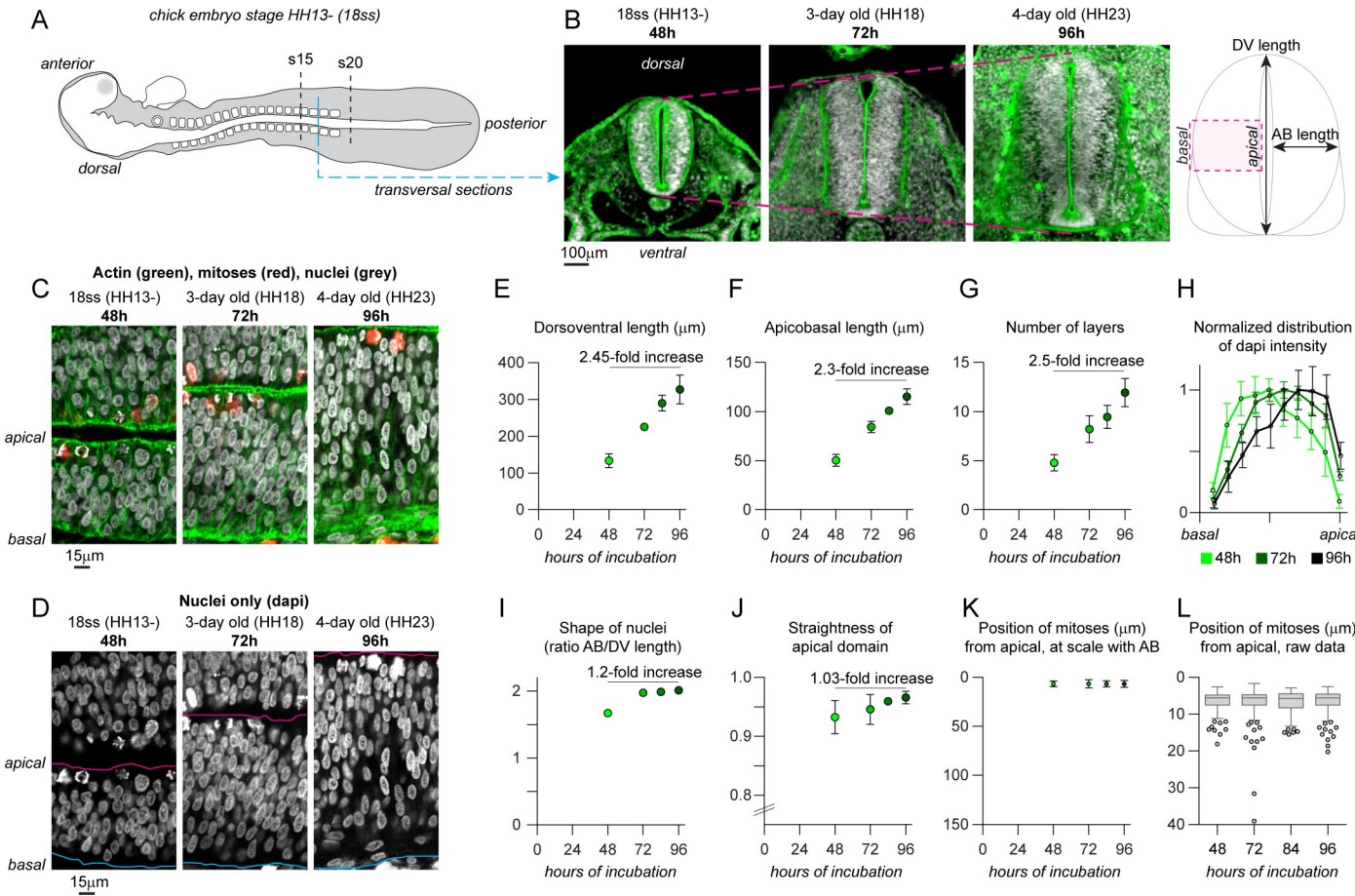

**Fig 1. Growth of caudal neuroepithelium of the chicken embryo from 2 to 4 days of development.** A. Diagram of a chicken embryo at 48h of incubation (stage HH13-, 18 somites). B, transversal cryosections at the level of the forelimb at 48h, 72 and 96h of incubation. C-D, Confocal images of the intermediate region of the neuroepithelium at the level of the forelimb at 48, 72 and 96h of incubation, nuclei are stained with DAPI (grey) and actin with Phalloidin (green). Magenta line, apical domain; cyan line, basal domain. Note that the basal region progressively becomes devoid of nuclei. E, net dorsoventral length of the neural tube. F, net apicobasal length of the neural tube. G, number of pseudolayers of nuclei along the apicobasal axis. H, distribution of DAPI intensity along the apicobasal axis, normalized to the peak intensity in each dataset and to the apicobasal size so that the various stages can be compared. I, ratio of apicobasal and dorsoventral length of nuclei. J, straightness of apical domain (net distance between dorsal-most and ventral-most points of the apical surface divided by the actual length of the apical surface between these two points). K, positions of mitotic cells (either phospho-histone H3-positive cells or cells with condensed chromosomes from DAPI staining), at scale with the actual apicobasal size of the neural tube. L, positions of mitoses, raw data. Descriptive data collected from 20 embryos. Dots represent mean values. Error bars show the standard deviation. Box and whiskers plot: the box extends from the 25th to the 75th percentile; the whiskers show the extent of the whole dataset. The median is plotted as a line inside the box. AB, apicobasal; DV, dorsoventral; HH, Hamburger-Hamilton stages of chicken development; ss, somites.

Importantly, we focused on the intermediate region (half way between the dorsal and ventral sides of the tissue) to avoid the effect of extensive neuronal delamination and differentiation occurring in the ventral region of the tissue at these stages. We then performed immunostaining for phospho-histone H3 to label cells in mitosis and counterstained for actin and DNA (Fig 1C and 1D). We found that the dorsoventral size of the neuroepithelium increases 2.45-fold over two days, going from an average of 133μm to 326μm (Fig 1E), while the apicobasal size of the tissue grows by 2.3-fold from 50μm to 115μm (Fig 1F). Over the same period, the antero-posterior distance from somite 15 caudalward increases 3.56-fold, going from 2.6 to 9.4mm (S1 Fig). This indicates that growth of the posterior neuroepithelium between 2 and 4 days of development is biased toward anteroposterior growth. The number of pseudolayers of nuclei from apical to basal increases by 2.5-fold (Fig 1G). Interestingly, the distribution of nuclei changes from a homogenous distribution along the apicobasal axis in 48h-old embryos

(Fig 1H, light green curve) to an accumulation of nuclei in the apical region in 72 and 96h-old embryos (Fig 1H, dark green curves). Note that nuclei density is lower in the basal region of the epithelium (Fig 1D). A change of average nuclear shape is also observed. The aspect ratio goes from 1.67 at 2 days to 2.01 at 4 days (Fig 1I). Nuclei become more elongated along the apicobasal axis (Fig 1D). This is due to a shortening of the nuclear length along the DV axis between 2 and 3 days of development while the length along the AB axis remains constant. By contrast, other parameters such as the straightness of the apical domain (Fig 1J), or the mean position of mitoses along the apicobasal axis (Fig 1K and 1L) do not significantly change (fold change inferior to 1.1). From these observations, we next wondered whether the balance between cell adhesion, proliferation and INM would be sufficient to drive the growth of the neuroepithelium, the progressive apical accumulation of nuclei and the increased pseudostratification, while apical positioning of mitoses and apical straightness remain constant.

## Agent-based model of PSE dynamics

To be able to assess the impact of INM versus other cytoskeleton-dependent processes (e.g. adhesion, mitosis), we built an agent-based model of the neuroepithelium. The chick neuroepithelium is an elongated PSE meaning that cells are very thin tubes with a large protruding nucleus giving them a watermelon-in-a-sock morphology (Fig 2A, S1 Movie). Cells are polarized according to the apicobasal axis with most of the cell-cell junctions localized apically and, conversely, the cell-matrix adhesions located basally [29]. In the model, each cell is approximated to a nucleus, an apical point and a basal point. The two points are attached to the nucleus via dynamic adjustable springs representing the viscoelastic properties of the cytoplasm (Fig 2B). Cells are placed next to one another along a lateral axis perpendicular to the apicobasal axis. Since the model is in 2D, this lateral axis can represent either the dorsoventral axis or the anteroposterior axis. Importantly, cells cannot intercalate nor swap positions. To model cell-cell and cell-matrix interactions, we use simple mechanical and behavioural rules. On the apical side, apical points are attached to each other by apical-apical springs representing cell-cell adhesion. On the basal side, basal points are attached to a fixed basal line representing the basal lamina of the epithelium. Basal points can only move along the basal line and a maximum distance between adjacent basal points is implemented to avoid uncontrolled flattening of the tissue along the basal line. As in the real epithelium, apical points are only attached to their direct neighbors and thus can move within the 2D domain. Neuroepithelial cells are known to keep a straight shape. Therefore, an alignment mechanism is set to prevent the apical point, the nucleus and the basal point of each cell from deviating significantly from a straight line. Nuclei cannot overlap. In the literature, non-overlapping constraints are approximated by a soft repulsion potential [30]. However, despite being computationally less expensive, this approximation becomes less and less accurate as the compression forces generated by congestion increase. Instead, we consider the nucleus being formed by an inner sphere (the hard core) and an outer sphere (the soft core). This representation allows soft cores to overlap with one another representing the deformation that would occur when two nuclei are pressed against each other [31]. In the chick neuroepithelium, nuclei are slightly compressed along the dorsoventral axis, giving them an elongated form along the apicobasal axis (see Fig 1C and 1D). Overlap of soft cores leads to a repulsive force. In addition, there is a non-overlapping constraint imposed to the nuclei hard cores.

A clock, representing a simplified cell cycle, rules the proliferation rate. This *in silico* cycle has 3 phases. A first phase corresponding to G1, S and the part of G2 during which no directed movements of nuclei take place (Fig 2C; G1/S/passive G2). A second phase accounting for the active nuclear movements occurring in G2 known as pre-mitotic rapid apical movements

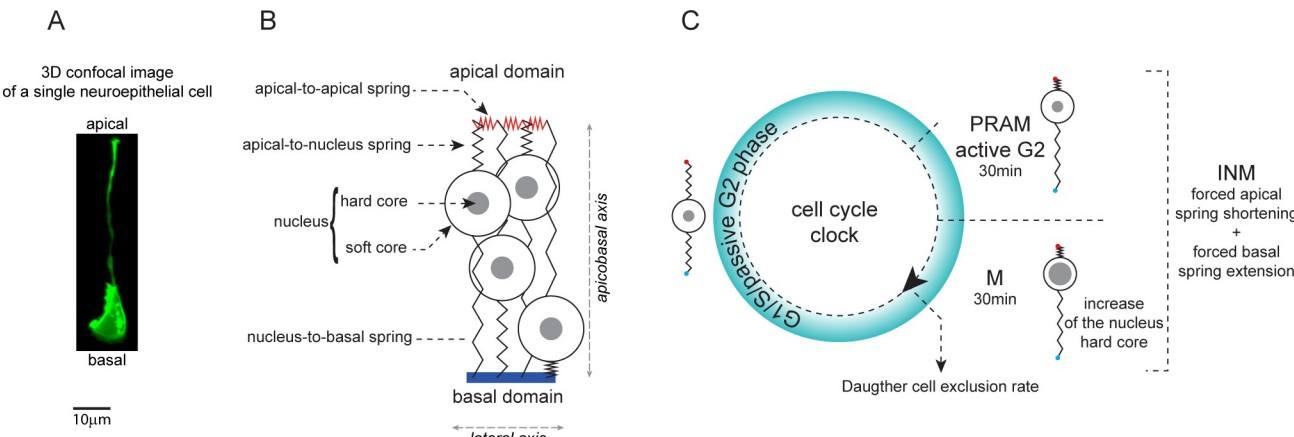

**Fig 2. Agent-based model of the pseudostratified epithelium.** A. 3D confocal image of a single neuroepithelial cells expressing membrane-GFP, in a 2-day old chick neuroepithelium (see S1 Movie). B. Cells in the model are abstracted to a nucleus attached to a set of springs. C, implementation of cell cycle and INM during the simulation. Cells in the model constantly proliferate by going through a simplified cell cycle corresponding to three phases: a G1/S/passive G2 phase during which springs connected to the nucleus adjust to local constraints, a PRAM/active G2 phase during which apical-nucleus springs shrink while nucleus-basal springs elongate to recapitulate INM movements and the M phase during which springs behave as in G2. In addition, the hard core of nuclei enlarges in M phase to account for cell swelling and stiffening. Finally, at the end of the M phase each cell gives two daughter cells. One is systematically kept within the 2D-plane, the other daughter cell can be excluded. This parameter allows to control the rate of growth of the tissue independently of the pace of the cell cycle. See S1 Information for a detailed description of the model.

(PRAM; Fig 2C, PRAM/active G2). A third phase representing Mitosis (Fig 2C; M). PRAM are implemented as follows: in cells entering active G2, the preferred rest length of apical-nucleus springs is set to 0 and the preferred rest length of the nucleus-basal spring is set to the current height of the cell. This drives an active apical-ward movement of the nuclei. In addition, during mitosis, the hard core of the nucleus increases (Fig 2C). Given that in our model cells have no cytoplasm, the increase of the hard core represents the known swelling and stiffening of cells during mitoses [32, 33]. At the end of mitosis daughter cells can be kept within the 2D-plane of the model or excluded. Thus, by systematically excluding 50% of the daughter cells we can keep the size of the cell population constant. This allows us to decouple the cell cycle and INM from actual proliferation (increase of cell number over time). Importantly, given the current lack of consensus about systematic active basal-ward movement of nuclei during G1 and S phases, we chose not to implement active movements of nuclei toward the basal side of the tissue in our model.

Outside of the PRAM/active G2 and M phases, apical-nucleus and nucleus-basal springs adjust their preferred rest length to their actual length, thus incorporating viscous behavior into the cytoplasm dynamics. This allows cells to accommodate their nuclei all along the apico-basal axis according to local constraints and forces (e.g., the position of the other nuclei or the forces on the various springs). Furthermore, there is a noise factor that allows nuclei to randomly move from their current location at each iteration of the simulation. In the chick neuroepithelium, nuclei are not known to display large scale random movements outside of PRAM. Consequently, in our simulations, the noise is set very low compared to PRAM.

Each simulation is initialized with 30 cells. All cells have their apical point, nucleus and basal point aligned. Apical points and basal points are evenly distributed. This can be seen in the first frame of S2–S6 Movies. Then, at each time-iteration, the simulation runs as follows: springs and nuclei are updated according to the position of each cell in the cell cycle, cells that are in mitosis divide and noise is implemented. Mitosis and noise may lead to the violation of the non-overlapping constraints on the nuclei hard cores. A minimization algorithm developed in [34] is then used to obtain an admissible configuration. This configuration

corresponds to a local minimizer of the total mechanical energy in the system associated to the springs, nuclei soft core and alignment forces. A complete description of the mathematical model can be found in S1 Information and all parameters used for the simulations presented in all figures hereafter are summarized in S1 Table.

## INM oppose apicobasal elongation, generate apical nuclear crowding and enlarge the apical domain

To start with, we checked the evolution of the tissue in absence of proliferation (no INM, Fig 3) to assess the influence of cell-cell, cell-matrix adhesions and the non-overlapping constraints between nuclei. To do so, we set the minimum duration of the cell cycle to 10000 hours making it unlikely that any cells would divide during the course of the 48-hour simulation. The apical-apical springs were set to be passive, meaning that they do not adjust their size in response to stretch or compression. In these conditions, there is no change in apicobasal size of the tissue (Fig 3A, red curve, AB) with nuclei distributing homogenously halfway along the apicobasal axis (Fig 3A, red curve, N). The number of pseudolayers remains constant (Fig 3B, red curve) and the apical domain stays flat (Fig 3C, red curve). See S2 Movie. In order to assess the effect of INM without adding more cells to the tissue, we set the exclusion rate of daughter cells to 50%. This means that after mitosis only one of the daughter cells is kept in the 2D-plane, keeping the total cell number constant. The total cell cycle duration is set to a range of 10 to 21 hours corresponding to averages of the known values for the duration of the cell cycle in the chick trunk neuroepithelium between 2 and 4 days of development [35]. G2 and M phases respectively last 90 and 30 minutes each, however active nuclear movements corresponding to PRAM only occur in a fraction of the total G2. Thus, in the model to generate normal INM conditions we set PRAM/active G2 and M phases to 30 min each (normal INM, Fig 3A–3C, black curves). To generate low INM conditions where cells actively displace their nuclei for a shorter period, we set the duration of PRAM/active G2 to 0 and M to 6 minutes (one iteration only) (low INM, Fig 3A–3C, brown curves). Introducing low or normal INM slightly reduces the apicobasal size of the tissue (Fig 3A, brown and black curves AB are below the corresponding red curve) and the average position of nuclei shifted apically (Fig 3A, brown and black curves N are above the corresponding red curve). Adding INM also leads to a slight decrease in terms of pseudolayers (Fig 3B, black curve) and apical straightness (Fig 3C, black curve). Normal INM parameters lead to apical mitosis whereas low INM lead to a widespread distribution of mitoses along the apicobasal axis (Fig 3D). These data indicate that INM are sufficient to drive global nuclear apical crowding (Fig 3A, black curve), to slightly destabilize apical straightness and of course to control apical positioning of mitoses. In addition, we checked the net lateral expansion of the apical, nuclear and basal domains under each of these 3 conditions. Each domain is defined by the length between the proximal and distal apical, nuclear and basal points along the lateral axis, respectively (Fig 3E–3G). These analyses reveal that INM promote the expansion of the apical domain, especially when all mitoses are apical (Fig 3E–3G, magenta curves, arrow). See S2 Movie.

Apical domains of epithelia are known to be dynamic and to display actomyosin contractility [36]. In addition, epithelial cells are known to resolve local imbalances in tension, compression and shear by aligning their cytoskeleton [37]. Further, apical contractility is known to be important for epithelial cell shape changes from squamous to columnar which corresponds to an apicobasal elongation [16, 36]. Thus, apical contractility may be an important driving force for apicobasal elongation in PSE. To explore this hypothesis, we first check that actomyosin contractility was important in the chick neuroepithelium by treating samples with the ROCK inhibitor, a compound specifically blocking Rho-dependent myosin contractility (S2 Fig).

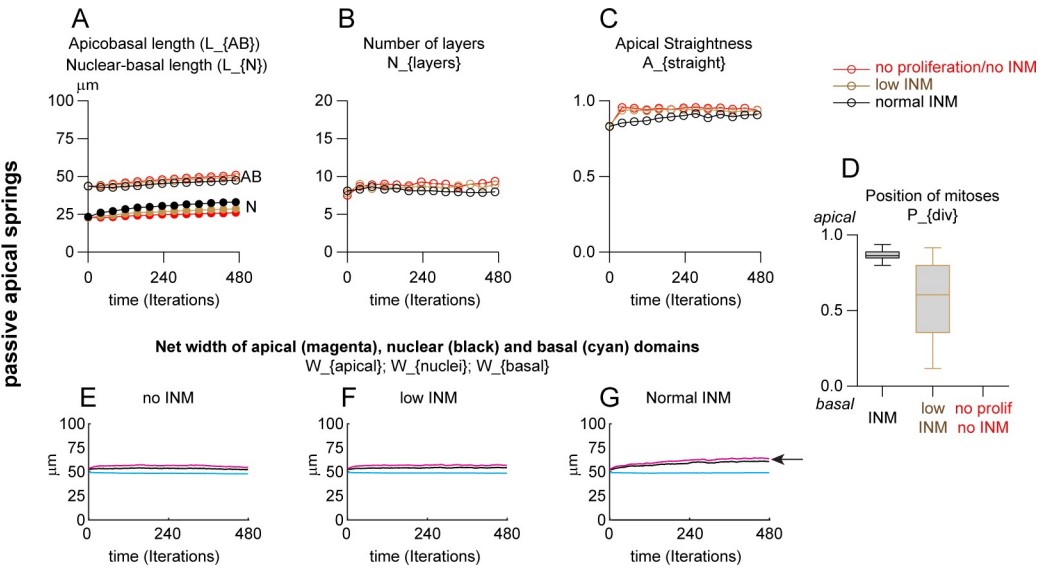

**Fig 3. INM opposes apicobasal elongation, generates apical nuclear crowding and enlarges the apical domain.**
Simulations with passive apical-apical springs with INM (black and brown curves) or without INM (red curves) with
constant cell number (see S2 Movie). A, Apicobasal length of the PSE (AB) and mean nuclear position along the AB axis,
expressed in micrometers. Note that INM reduce apicobasal length (black curve with open circles (INM) is below the red
curve (no INM)) and generate apical nuclear crowding (black curve with closed circles (INM) is above the red curve (no
INM)). B, Number of layers of nuclei along the AB axis. C, straightness of apical domain (net distance between the first
and last apical point divided by the actual distance between these two points). D, Position of mitoses along AB axis, 1 being
apical. E-F, net width of apical (magenta), nuclear (black) and basal (cyan) domains of the PSE over time. For each
domain, the distance the between the first and last point along the lateral axis is computed and its evolution plotted over
time. Note that INM promote enlargement of the apical domain (panel 3G, arrow). Each simulation was performed over
480 iterations (48h of biological time) for 10 repetitions. Each curve represents the mean value of each dataset for the
parameter plotted. Box and whiskers plot: the box extends from the 25th to the 75th percentile; the whiskers show the
extent of the whole dataset. The median is plotted as a line inside the box.

Indeed, a brief 2-hour treatment with the ROCK inhibitor leads to a decrease of the apicobasal
length, a reduction in pseudostratification, a rounding of nuclei and a decrease of the apical
straightness. In addition, mitoses are not systematically apical (S2 Fig). Importantly, the
nuclear staining using DAPI indicates that the tissue is healthy as no fragmented nuclei were
observed. To further confirm that exposure to the ROCK inhibitor did not trigger massive
cell death during the course of the experiment, we repeated the treatment and performed a
TUNEL assay to detect DNA degradation (S3 Fig). Interestingly, by contrast to inhibiting acto-
myosin contractility, blocking actin polymerization using a Rac1 inhibitor only had a mild
effect on all these parameters (S2 Fig). These data indicate that the chick neuroepithelium has
a short-term reliance on actomyosin contractility to maintain its shape and mitoses positions.
Therefore, we ran the same simulations as above but setting the preferred rest length of apical
springs to 0 in order to model overall apical contractility. Therefore, apical springs dynamically
adjust their sizes to reach this rest length (Fig 4). We found that introducing contractile apical
springs leads to a slight increase of apicobasal elongation (Fig 4A, AB curves) but overall
nuclear positioning (Fig 4A, N curves), pseudostratification (Fig 4B) and tissue shape (Fig 4C)
are not dramatically affected by having a contractile apical domain. In the absence of INM,
contractile apical springs lead to a rapid shrinkage of the apical domain (Fig 4E, arrowhead).
This effect is prevented by adding INM (Fig 4G, arrow) indicating that apical mitoses can
oppose apical contractility (S2 Movie).

Apart from being attached to one another, cells in PSE are also strongly attached to the
extracellular matrix by their basal domain. We wondered, whether affecting cell-matrix

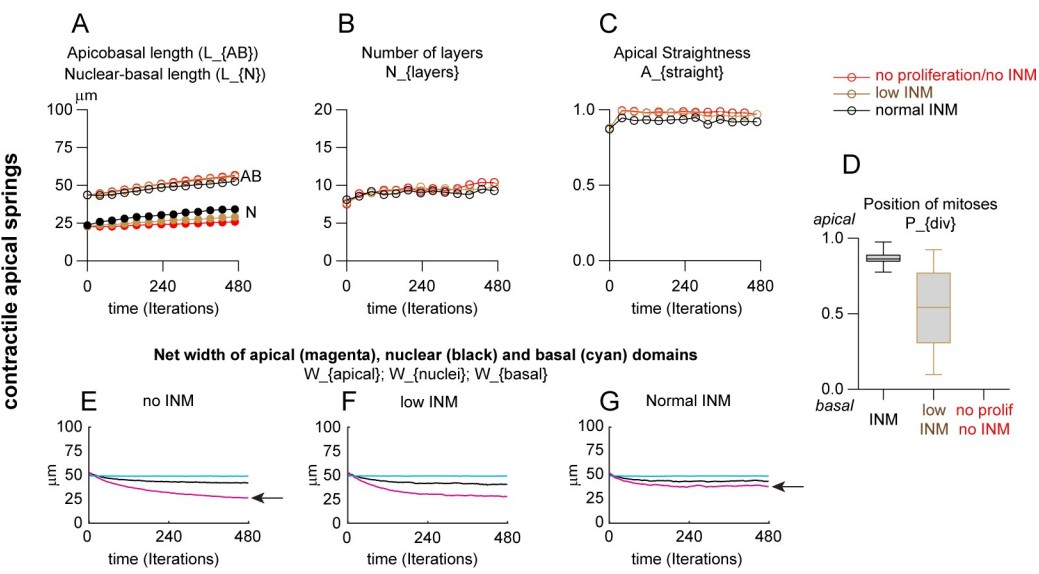

**Fig 4. INM opposes apical contractility.** Same simulations as in Fig 3 but with contractile apical-apical springs with INM (black and brown curves) or without INM (red curve), with constant cell number (see S2 Movie). A, Apicobasal length of the PSE (AB) and mean nuclear position along the AB axis, expressed in micrometers. B, Number of layers of nuclei along the AB axis. C, straightness of apical domain (net distance between the first and last apical point divided by the actual distance between these two points). D, Position of mitoses along AB axis, 1 being apical. E-G, net width of apical (magenta), nuclear (black) and basal (cyan) domains of the PSE over time. For each domain, the distance between the first and last point along the lateral axis is computed and its evolution plotted over time. Note that apical contractility reduces the width of the apical domain (panel 4E, arrow), whereas introducing INM opposes apical contractility (panel 4G, arrow; red curve in G is higher than in E). Each simulation was performed over 480 iterations (48h of biological time) for 10 repetitions. Each curve represents the mean value of each dataset for the parameter plotted. Box and whiskers plot: the box extends from the 25th to the 75th percentile; the whiskers show the extent of the whole dataset. The median is plotted as a line inside the box.

adhesion may lead to defects of nuclear, cell or tissue shape. To do so, we treated trunk explants with Dispase, an enzyme capable of degrading collagens and fibronectin (S4 Fig). Interestingly, decreasing cell-matrix adhesion led to a reduction of apicobasal size, nuclei rounding, a decrease of pseudostratification, impaired apical straightness and to some non-apical mitoses. These data indicate that the overall balance of tension across both cell-cell and cell-matrix adhesion is important for the shape of the chick PSE. Impairing cell-matrix adhesion sometimes triggered local apical extrusion. Importantly, all defects in nuclei and tissue shape described above were also detected in regions where no apical extrusion had taken place suggesting that the tissue relaxation occurs first and then some cells may extrude apically as a consequence of the loss of cell-matrix adhesion.

## Increase in cell number strongly increases pseudostratification but has a weak effect on apicobasal elongation

Since neither INM nor apical contractility are sufficient to drive extensive apicobasal elongation, we next wanted to compare the impact of having passive or contractile apical-apical springs in the context of increasing cell number by proliferation (allowing more than one daughter cell to remain in the 2D plane after mitosis). For that, we ran two sets of simulations with passive or contractile apical-apical springs and with different rates of exclusion of daughter cells ranging from 0% (all cells generated by mitosis are added to the 2D plane) to 50% (one daughter cell is systematically excluded). INM are set to normal with PRAM/active G2 and M phases lasting 30 min each in all conditions. All outputs of simulations with passive apical-

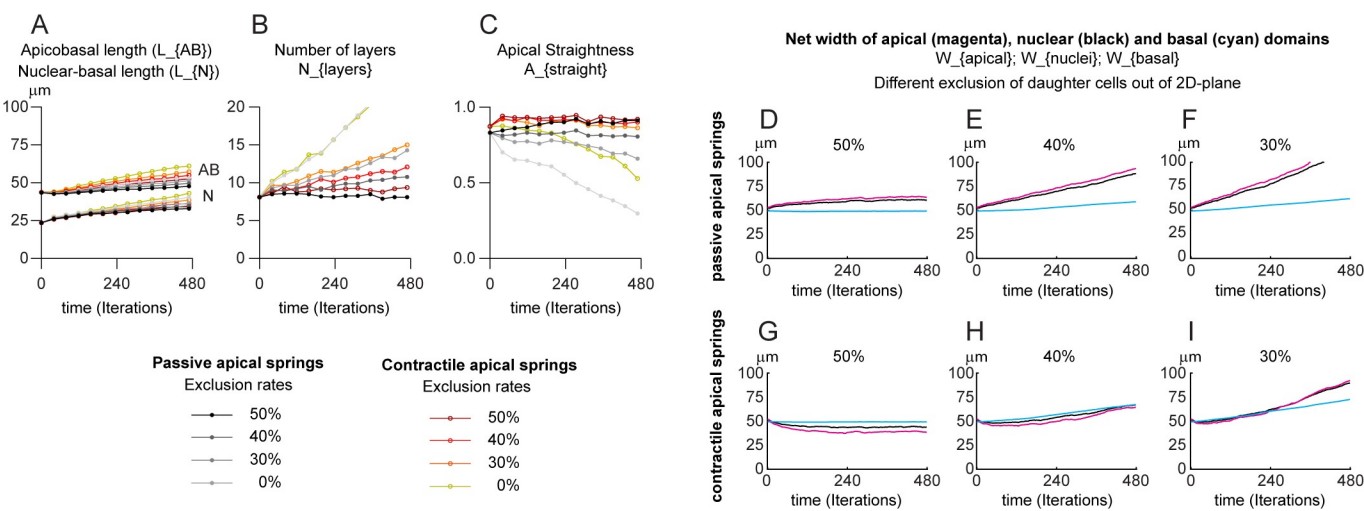

**Fig 5. Increase in cell number drives pseudostratification and apical contractility feeds back into basal rearrangements.** Simulations with passive or contractile apical springs, normal INM and various rates of exclusion of daughter cells (see S3 Movie). A, apicobasal length of the PSE (AB) and mean nuclear position along the AB axis (N) over time expressed in micrometers. B, Number of pseudolayers of nuclei along the AB axis. C, straightness of apical domain (net distance between the first and last apical point divided by the actual distance between these two points). D-F, net width of apical (magenta), nuclear (black) and basal (cyan) domains of the PSE with passive apical-apical springs over time with 50 (D), 40 (E) and 30% (F) of daughter cells being excluded from the 2D plane. G-I, net width of apical (magenta), nuclear (black) and basal (cyan) domains of the PSE with contractile apical-apical springs over time with 50 (G), 40 (H) and 30% (I) of daughter cells being excluded from the 2D plane. For each domain, the distance between the first and last point along the lateral axis is computed and its evolution plotted over time. Note that apical contractility leads to basal rearrangements (compare cyan curves in H-I grow faster than in E-F). Each simulation was performed over 480 iterations (48h of biological time) for 10 repetitions. Each curve represents the mean value of each dataset for the parameter plotted.

apical springs are plotted in shades of grey to black whereas outputs for simulations with contractile apical-apical springs are plotted in shades of hot colors from yellow to dark red. Contractile apical-apical springs have a slight positive impact on tissue apicobasal elongation (Fig 5A, AB curves; all colored curves are above their corresponding grey-to-black curves). This is accompanied by a slight apical shift of nuclei (Fig 5A, N curves). Pseudostratification correlates directly with the number of cells retained in the 2D plane (Fig 5B). At maximal (50%) or intermediate (40%, 30%) exclusion rates of daughter cells, contractile apical-apical springs further increase pseudostratification (Fig 5B, orange and red curves are above their cognate grey-to-black curves; S3 Movie). This effect is lost when all cells are retained (Fig 5B, 0% exclusion, yellow and grey curves overlap). With passive apical-apical springs, the PSE shape becomes very sensitive to an increase of cell number. The more cells are kept within the 2D plane of the epithelium the faster apical straightness decreases (Fig 5C). Introducing contractile apical-apical springs mitigates the effect of hyper-proliferation (Fig 5C, orange and red curves stay close to straightness of 0.9). We then monitored the net lateral expansion of apical, nuclear and basal domains over time for all conditions (Fig 5D–5I). An increase in cell number induces a rapid expansion of the apical domain (Fig 5D–5F). Interestingly, the apical shrinkage induced by contractile apical springs (Fig 5G, red curve) can be opposed by increasing the number of cells (Fig 5H and 5I, magenta curves). In addition, apical contractility positively feeds back into basal expansion. Note that the cyan curves in panels 5H and 5I increase faster than in panels 5E and 5F. These data indicate that an increase in total cell number drives a slight increase in apicobasal length, strongly drives pseudostratification and, in addition to INM, leads to an expansion of the apical domain. Interestingly, in the context of increasing cell number, apical contractility promotes basal rearrangements, an effect not seen with constant cell numbers (compare the cyan curves in panels 3E-G and 4E-G, with cyan curves in panel 5D, 5G). Further, apicobasal constriction slightly contributes to pseudostratification and helps maintain tissue shape during tissue growth.

In vivo, the chick neuroepithelium is a closed tube and is surrounded by other structures (e.g. superficial ectoderm, paraxial mesoderm). It experiences physical constraints that may contribute to its overall shape. To assess whether restricting lateral expansion of our growing simulated PSE would help maintain tissue shape, we ran simulations with passive apical springs and different exclusion rates of daughter cells as in Fig 5 and added lateral walls (S5 Fig, S4 Movie). With constant cell number (exclusion rate of daughter cells at 50%) the straightness of the apical surface is low but constant. When including cell proliferation (exclusion rates of 40 and 30%), intense buckling of the apical surface is observed, indicating that adding lateral physical constraints do not contribute to maintaining tissue shape during tissue growth contrary to what apical contractility does.

### Neuroepithelial cells undergo a dramatic change in shape that exceeds what is needed to accommodate nuclei along the apicobasal axis

So far, our simulations reveal that a progressive increase of pseudostratification, apical nuclear accumulation and apical mitoses can emerge from cell-cell/cell-matrix adhesion, proliferation and INM. However, none of the conditions tested allows a rapid apicobasal elongation of the tissue over 48h. This suggests that something is missing in our model. To achieve fast apicobasal elongation, cells could either get bigger (increase of cell volume) or elongate beyond what is necessary to accommodate the increase in number of nuclei due to proliferation. Our descriptive in vivo data (Fig 1A and 1B) show that a region of low nuclei density is formed in the basal domain of the chick neuroepithelium between 48 to 96 hours of incubation. This is driven by INM in our model (Fig 3A). Such low basal density of nuclei has been seen in other elongated PSE as well [29]. This observation suggests that apicobasal elongation of cells may not be driven by the pilling of nuclei along the apicobasal axis. Could elongation be caused by a change of cell shape or are cells also changing in size? To answer this question, we dissected explants of the neuroepithelium from the forelimb region (facing somites 15 to 20) in embryos at 2, 3 and 4 days of development (Fig 6A). Neural tubes were enzymatically isolated from surrounding tissues and dissociated to produce a suspension of single neuroepithelial cells (see material and methods). Cells were automatically counted and their diameters retrieved using a cell counter. From these measurements, volumes were calculated. This analysis reveals that the mean volume of neuroepithelial cells decreases between 2 and 3 days of development and remains stable from 3 to 4 days (Fig 6B and 6C). We next checked the size of the apical domains by performing *en face* microscopy on neuroepithelia from the same stages (Fig 6A, 6D and 6E). We found that the mean area occupied by the cells' apical sides is also significantly getting smaller between 2 and 3 days of development but does not significantly change from 3 to 4 days (Fig 6D and 6E). These data indicate that neuroepithelial cells undergo a dramatic change of cell shape, together with a reduction of cell volume, which appear to exceed what would be needed to accommodate the increase of cell nuclei along the apicobasal axis. This strongly suggests that undifferentiated PSE cells specifically elongate rather than simply adjust to local nuclear crowding.

### Apicobasal elongation requires a specific elongation force

Therefore, we wondered whether adding a global non-oriented expansion force might account for the apicobasal elongation observed in vivo. To do that, we increased the amount of noise on nuclear position at each iteration. All nuclei are allowed to move in a random direction at each iteration of the simulation. Since apical-nucleus and nucleus-basal springs are able to update their rest length to adjust to their actual size, any increase in random nuclear movements forces the apical-nucleus and nucleus-basal springs to stretch. Given that cells are

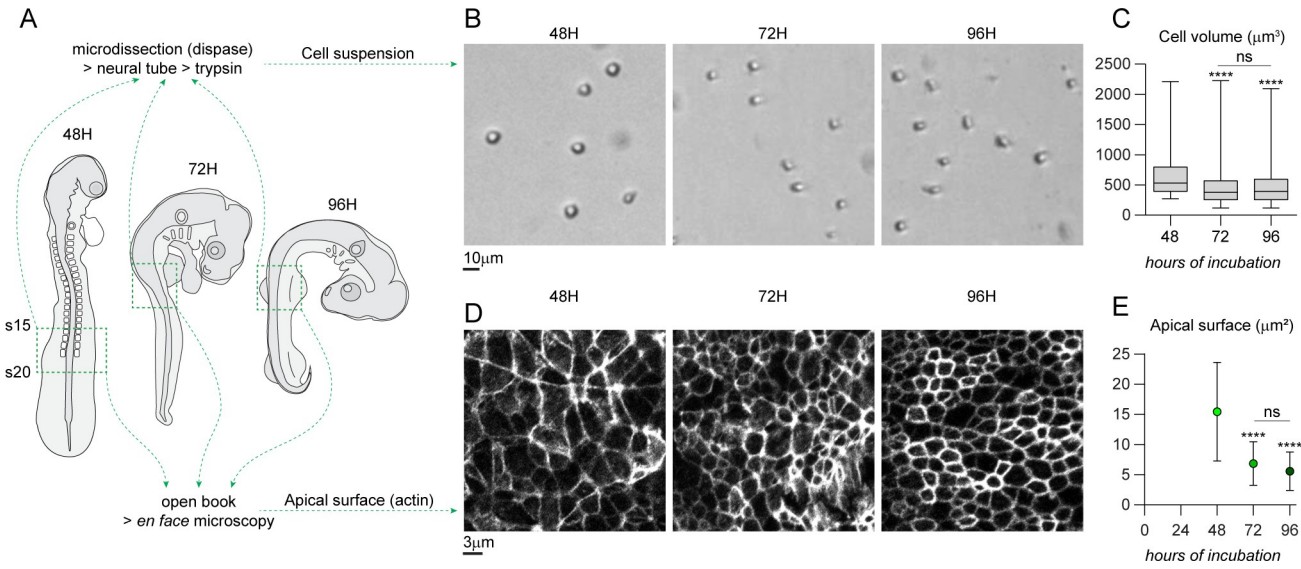

**Fig 6. Mean volume and mean apical surface of chick neuroepithelial cells decreases between 2 and 3 days of development.** A, Diagram depicting the regions used for the preparation of cell suspensions and open book histology from chicken embryos at 2, 3 and 4 days of development (see Methods for details of the experimental procedures). Region monitored (from somite 15 to 20, forelimb region) is indicated by a dotted line. B, representative images of neuroepithelial cells in suspension after neural tube dissection and enzymatic dissociation. C, mean volume of neuroepithelial cells over time ($n_{48h} = 82$; $n_{72h} = 568$; $n_{96h} = 1168$). Cells get significantly smaller from 2 to 3 days of development and remain stable. Box and whiskers plot: the box extends from the 25th to the 75th percentile; the whiskers show the extent of the whole dataset. The median is plotted as a line inside the box. One-way ANOVA (Kruskal-Wallis) followed by Dunn's multiple comparisons. ****, p<0.0001; ns, p>0.9999. D, *en face* view of the apical domain of the intermediate region of the neuroepithelium (actin is stained by Phalloidin). E, mean area of the individual apical surfaces over time ($n_{48h} = 67$; $n_{72h} = 66$; $n_{96h} = 81$). Apical surfaces shrink from 2 to 3 days of development and remain stable. Dots represent mean of the dataset, error bars represent S.D. One-way ANOVA followed by multiple comparisons. ****, p<0.0001; ns, p = 0.1586.

attached to the basal line, to each other and that cells are prevented from bending due to an imposed alignment force, increasing random nuclear noise should generate a linear apicobasal elongation force. We ran simulations with conditions similar to those presented in Fig 4 but with a 25-fold increase of random nuclear movements (Fig 7). Under these conditions, we observed a 2-fold increase in apicobasal length (Fig 7A, S5 Movie) whereas the number of pseudolayers (Fig 7B) and apical straightness (Fig 7C) were similar to the values obtained with low noise (Fig 5B and 5C). In addition, increasing noise does not affect the overall dynamics of lateral expansion of the tissue (Fig 7D–7F). At low percentages of exclusion of daughter cells (30%) the lateral expansion of the apical domain is faster than that of the basal domain (Fig 7F, compare magenta and cyan curves). We attempted to solve this issue by allowing basal points to update their positions at a faster rate. This was sufficient to allow an isotropic expansion of the tissue (Fig 7G, black, cyan and magenta curves grow at the same pace; S5 Movie). Finally, our previous simulations (Figs 3 and 4) hinted that INM were capable of opposing apicobasal elongation. We wanted to check if this was still true under the extensive apicobasal growth generated by increased noise. We repeated the same simulations as shown in Fig 7, excluding 50% of daughter cells to keep total cell number constant, but setting low INM conditions with PRAM/active G2 set to 0 and M to 6min. (S6 Fig). Under low INM conditions, the PSE elongates along the apicobasal axis faster than with normal INM and there is a shrinkage of the apical domain (S6 Movie) confirming our previous observations.

In conclusion, our data indicate that: i/ pseudostratification is mainly controlled by the increase of cell number, ii/ apical contractility is essential to maintain tissue shape in the context of a high proliferation rate, iii/ INM promote the expansion of the apical domain, iv/ INM oppose pseudostratification, apical contractility and apicobasal elongation whereas they

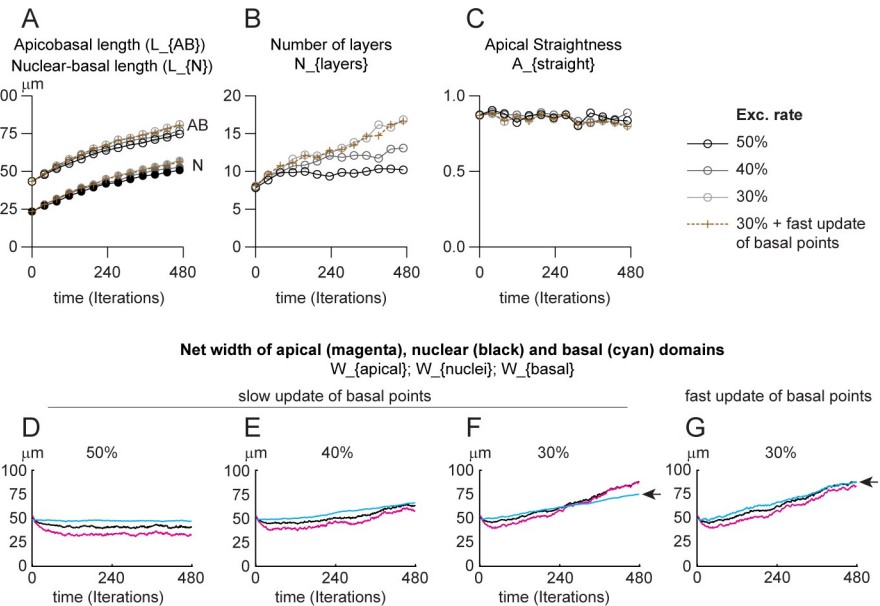

**Fig 7. Apicobasal elongation requires a specific elongation force.** Simulations with contractile apical springs, normal INM and high noise. A, apicobasal length of the PSE (AB) and mean nuclear position along the AB axis (N) over time expressed in micrometers (see S5 Movie). B, Number of pseudolayers of nuclei along the AB axis. C, straightness of apical domain (net distance between the first and last apical point divided by the actual distance between these two points). D-G, net width of apical (magenta), nuclear (black) and basal (cyan) domains of the PSE with dynamic apical-apical springs over time with 50 (D), 40 (E) and 30% (F-G) of daughter cells being excluded from the 2D plane. For each domain, the distance between the first and last point along the lateral axis is computed and its evolution plotted over time. D-F, slow update of basal point positions (as in all previous Figs). G, fast update of basal point positions (twice faster). Each simulation was performed over 480 iterations (48h of biological time) for 10 repetitions. Each curve represents the mean value of each dataset for the parameter plotted.

control apical positioning of mitosis and apical nuclear crowding, v/ apicobasal elongation of cells is likely to be due to an active elongation program and not a mere consequence of increased nuclear density (Fig 8).

## Discussion

Our simulations clearly link INM to the rapid emergence of nuclear crowding in the apical domain and to the formation of a region of low nuclear density in the basal part of the PSE. Yet, in vivo, the low nuclear density observed in the basal domain of the chick trunk neuroepithelium (forelimb level) only emerges in 3-to-4-day old embryos. This could be due to a lack of INM at early stages of neural plate/tube development. This is unlikely since apical mitoses have been observed even at open neural plate stages in chick and mouse embryos [38, 39]. Alternatively, at early stages, there could be a counterbalancing post-mitotic rapid apical removal (PRAR, Fig 8) of nuclei as part of the INM, as suggested in rats and ferrets' brains [7, 11]. Such active basal-ward movement would prevent the early formation of a crowded apical domain and that of a relatively loose basal domain. Neuroepithelial cells always detach from the apical surface upon differentiation into neurons but they can also be induced to detach from the apical surface by a local increase of the apical density of nuclei [40, 41]. Thus, it may be important to delay apical nuclear crowding to prevent early delamination of undifferentiated neural progenitors from the apical domain. Therefore, one could propose that regulation of the intensity of INM might control the onset of neuron delamination in order to synchronize neuronal differentiation with the development of the spinal cord itself. The loose basal

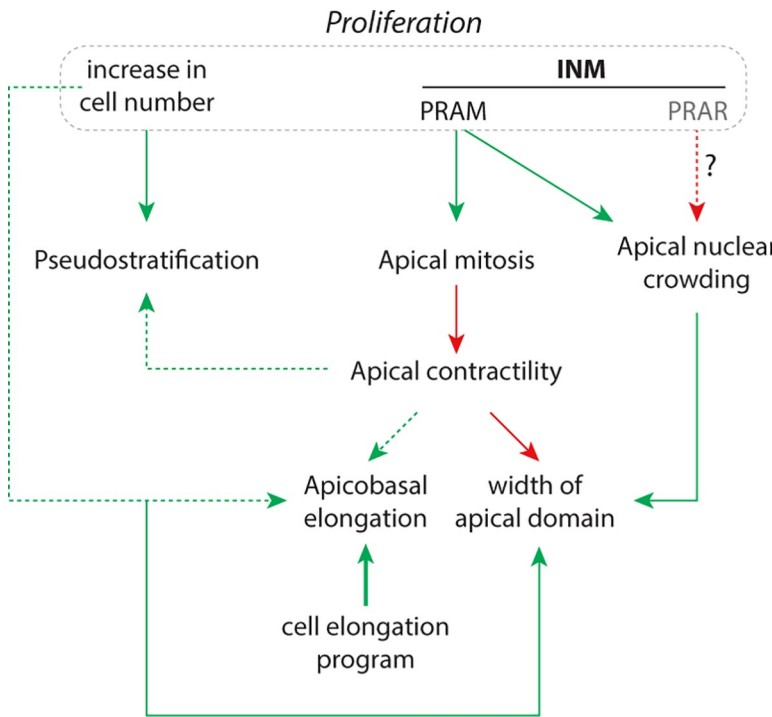

**Fig 8. Interplay between INM, proliferation and cell adhesion in the context of a specific apicobasal elongation program are needed to recapitulate normal PSE dynamics.** Green arrows indicate positive action, red arrows indicate negative/inhibitory action. Dotted line indicate weaker effect than plain lines. PRAM, pre-mitotic rapid apical migration; PRAR, post-mitotic rapid apical removal; INM, interkinetic nuclear movement.

region may also be needed to accommodate the cell body of delaminating neurons which accumulate in the basal side of the epithelium. In addition, or alternatively, INM might control the onset of neuron delamination in order to synchronize neuronal differentiation with the development of adjacent structures awaiting innervation. For instance, it would be interesting to see if somites (or the myotome), which produce muscles to be innervated, interfere with the patterns of INM in the adjacent neural tube.

Apical contractility is known to be essential to promote the apicobasal elongation driving the switch from squamous to cuboidal epithelial sheets [36]. Yet, in our simulation apical contractility was a poor driver of apicobasal elongation. This is likely due to the fact that in elongated PSE the apical surface is small compared to the size of the basolateral membrane of cells. In the trunk of a 2-day old chicken embryo, the diameter of the apical surface is around 5 microns whereas cells are already 50 microns tall. Thus, any change of the apical size will only have a marginal effect on cell height. At early stages, during the folding of the neural plate, actomyosin contractility is important and Rho and myosin are detected in the apical region of most neuroepithelial cells [42]. It is proposed that apical contractility may drive cell shape changes contributing to neural plate bending. However, the region that undergoes the most dramatic change of shape, the medial hinge point located at the midline above the notochord, has little and only transient accumulation of Rho. In addition, in mice with targeted defects in cytoskeletal genes, neurulation of the caudal neural tube is rarely affected [43]. Following observations in chick [44] and mouse [38, 45] embryos, it was alternatively proposed, that the change in cell shape at the medial hinge point could be driven by local differences in INM in the neural plate, with cells above the notochord having longer S-phase. The nuclei would spend more time in the basal regions favoring an enlargement of the basal domain. This would

let cells adopt a pyramidal shape promoting tissue folding. As for the effect of apical contractility, a longer S-phase is unlikely to generate any significant change in cell shape in elongated PSE since the width of the nucleus only represent a small fraction of the total cell height. It is interesting to note that while the intermediate region of the neuroepithelium exhibits a fast linear increase in apicobasal size from 2 to 4 days of development, the cells located in the floor plate of the neural tube do not change significantly in height during this period. If indeed their pyramidal shape is linked to a specific cell cycle with a long S-phase, this relationship can only be maintained if the cells do not elongate beyond a size that would mitigate the effect of the position of the nucleus. Furthermore, some PSE, such as the retina, exhibit an extreme curvature and cells are relatively short along the apicobasal axis. It would be interesting to explore the putative contribution of specific INM patterns to the cell and tissue shape. In particular, it would be interesting to see whether a longer S-phase or active basal-ward movements might contribute to cell wedging and tissue folding together with apical contractility or other actin based mechanism such as the actomyosin-dependent apicobasal pulling that has been described in the drosophila leg disc [46].

Histological analyses in mice with CAKUT (congenital anomalies of the kidney and urinary tract) or EA/TEF (Esophageal atresia/tracheoesophageal fistula) [2, 3] show a correlation between tissue expansion defects and local lack of INM. The local loss of INM (basal mitoses) correlates with problems of convergence extension and defects in tissue separation which were proposed to be due to a loss of apicobasal polarity in these syndromes. Interestingly, INM is not a permanent feature of all PSE. In mouse, INM stops around E13 in the esophagus meaning that from this stage basal mitoses naturally occur in this tissue [2]. By contrast, by E14, INM and apical mitoses can still be observed in the brain, the trachea, the ureter and the intestine. As with the correlation between lack of expansion and lack of INM in the aforementioned syndromes, it was proposed that INM might be a driving force of tissue expansion, since the organs in which it persists expand at a faster rate than the overall growth of the embryo at these stages. Here we show that INM directly contributes to expansion of the apical domain and thus promotes growth in axes perpendicular to the apicobasal axis (DV, AP). In vivo, other mechanisms likely contribute to tissue expansion along the dorsoventral or anteroposterior axis. One possibility is cell intercalation. This has been observed during neurulation in mouse [38]. The dorsal regions of the neural plate grow faster than the ventral regions. This is due in part to a faster cell cycle but also to significant cell intercalation from ventral to dorsal.

Another mechanism that could contribute to tissue expansion in one direction is the orientation of mitotic spindles. It was observed in the chick neural tube that, at early stages (HH7 to HH12, corresponding to 24 to 44h of incubation), most mitoses (circa 55%) were oriented along the anteroposterior axis [39]. Such percentages would correspond to an exclusion rate of daughter cells of 27% in our simulation. Therefore, with such a biased orientation, proliferation would favor anteroposterior elongation of the neuroepithelium compared to expansion in apicobasal and dorsoventral orientations. In addition, in most PSE studied, the mean cell cycle length tends to change over time. It can get longer as in the caudal neural tube [35] and in the brain [47] or shorter as in the ureteric epithelium [3] and the midgut [48]. Changes in cell cycle length will obviously change the rate at which a tissue increases in terms of cell number but will also affect the frequency of INM events. Given that apical localization of mitoses drives expansion of the apical domain by opposing apical contractility, such changes in cell cycle pace may also affect tissue shape. Indeed, in our simulations, imbalances in proliferation and apical contractility were sufficient to promote either bending or buckling of the apical surface (see S5 Movie). To maintain tissue shape and straightness of the apical domain, the rapid expansion of the apical domain needs to be compensated by an equivalent expansion of the basal domain or hindered by extensive apical contractility.

Further, defects in microtubule dynamics can lead to lissencephaly and microcephaly two common neurodevelopmental defects due to improper growth of the brain. This prompted some to suggest that impaired INM, a microtubule-dependent phenomenon, might contribute to these pathologies [49, 50]. However, all of these microtubule-related defects have also problems in neuronal migration, mitotic spindle positioning and proliferation making it difficult to identify the effects specifically due to a lack of INM.

Finally, our data indicate that apicobasal elongation is not a consequence of nuclear crowding linked to the increasing pseudostratification but is likely to require a specific cell elongation program. This is corroborated by the observation that during *Xenopus* neurulation the neuroepithelium thickens while not being pseudostratified [51], indicating that indeed lengthening along the apicobasal axis and pseudostratification are uncoupled. In the model, we generated the elongation force using an artificially increased nuclear noise. Given that, outside of PRAM, such large scale random nuclear movements were never observed in the chick neuroepithelium, it is very unlikely that in vivo apicobasal elongation comes from a progressive increase in random nuclear movements. Most likely, it comes from an extensive reorganization of the cytoskeleton. Experimentally disentangling the various putative cytoskeleton-related mechanisms involved in INM (PRAM or PRAR) and cell elongation will require the generation of new tools for the fine-tuning of specific subsets of actin/tubulin dynamics over long periods of time (hours to days) without affecting proliferation.

## Materials and methods

### Ethics statement

This research only used chicken embryos at early stages of development (before the 6th day). None of the procedures fall under legal requirements for animal use and can be performed by anyone without animal licence.

### Chicken eggs

Fertilized chicken eggs were obtained from S.C.A.L (Société Commerciale Avicole du Languedoc) and incubated at 38˚C until the desired stage [28].

### Enzymatic and drug treatments

10-somite long portion of the trunk at the level of the prospective forelimb region were dissected from embryos at stage HH12. Explants were then cultured in suspension in DMEM for 2 hours at 37˚C with Rock inhibitor (Y27632), Rac1 inhibitor (NSC 23766) at 400 μM, 1/250 dilution from 100mM stock solution or Dispase II (Stem Cell Technologies; #07923, at 0.2U/mL). All drugs are dissolved in DMSO, therefore control conditions for the drugs correspond to DMSO only at 0.4% in DMEM.

### TUNEL assay

Trunk explants were dissected and incubated in suspension under control, DMSO or Rock inhibitor conditions, fixed in formaldehyde 4% in PBS1X, washed in PBS 0.1% Tween and incubated overnight with the terminal transferase (Invitrogen, dTT 1730750) and a Dig-UTP nucleotide mix (Roche, 11093088910). The Dig-UTP was then detected using and anti-Dig coupled with alkaline phosphatase using NBT/BCIP as a substrate to generate a purple precipitate at the site of DNA degradation.

## Histology and staining of chicken embryo samples

Fixed embryos were soaked in Phosphate Buffer 15% sucrose overnight at 4°C. Then, embryos were transferred for 2 hours in gelatin 7.5%/ sucrose 15%. Small weighing boats are used as molds. A small layer of gelatin/sucrose is deposited at the bottom and left to set. Embryos are then transferred on the gelatin layer using a 2.5mL plastic pipette. Each embryo is placed in a single drop and left to set. Once all drops are set, an excess of gelatin/sucrose solution is poured on to the weighing boat to fill it. Once again gelatin is left to set on the bench. After setting, the dish is placed at 4°C for 1 hour to harden the gelatin. Once ready, the block of gelatin containing the embryos is placed under a dissecting microscope and individual blocks are carved to position the embryos in the desired orientation for sectioning. Cryosections were performed using a Leica CM1950 cryostat. Sections were incubated in PBS1X at 42°C for 30 minutes to remove the gelatin, treated with PBS1X, 1% triton, 2% serum for permeabilization and blocking. Primary antibodies were diluted in PBS1X 2% newborn calf serum and applied overnight at 4°C under a coverslip. Secondary antibodies were diluted in PBS1X and applied for 2 hours at room temperature. Washes were done in PBS1X. Antibodies used: mouse anti-phospho-histone 3 (Cell Signaling, MA312B). Counterstaining for actin and DNA was done with Phalloidin (1/1000) and DAPI (1/1000).

## Suspension of neuroepithelial cells

Samples of the whole trunk between somite 15 and 20 were taken from embryos at 2, 3 and 4 days of development. The explants were incubated in Dispase II (Stem Cell Technologies; #07923, at 1U/mL in DMEM) at 37°C for 20 minutes to degrade collagens and fibronectin. Tissues were then separated using mounted needles. Neural tube explants were then incubated in a trypsin solution (ThermoFisher, 25300054) to generate single cells. From neural tube explants from 3-day and 4-day old embryos, numerous cells (most likely neurons) did not adopt a round morphology after dissociation, instead they maintained an elongated fiber-like morphology and accumulated at the bottom of the tubes. They were not included in the supernatant used for cell diameter analysis.

## Open book observation

Samples of the whole trunk between somite 15 and 20 were taken from embryos at 2, 3 and 4 days of development. The neural tube was open from its dorsal side using forceps. The tip of one forceps is inserted in the neural tube lumen and moved along the anteroposterior axis to open the whole explant. Explants are then squeezed in between two coverslips to maintain them open. Apical side is positioned face down on an inverted microscope for observation.

## Imaging

Confocal images were taken on a Zeiss 710 confocal microscope. Whole mount images were acquired on a Leica MZ10F.

## Statistics

Statistical analyses of in vivo data were performed with Prism 6 (GraphPad). Datasets were tested for Gaussian distribution. Student t-tests or ANOVA followed by multiple comparisons were used with the appropriate parameters depending on the Gaussian vs non-Gaussian characteristics of the data distribution. Significance threshold was set at $p < 0.05$.

### In silico simulations and associated plots

The code was written in Fortran90 in sequential mode and the simulations were performed on a DELL Precision T7810 with windows 8.1, 64 bits of RAM, with two CPU Intel Xeon E52637 3.8 GHZ processors. The computational time of each simulation of the tissue evolution for 48 hours (480 time iterations) ranges from 8 minutes with 30 cells and no proliferation to approximately 40 minutes with proliferation and the exclusion rate of daughter cells set to 0%. For each set of in silico conditions at least 10 repetitions were performed. Data were processed using MatLab R2017b. Plots: for simplicity, mean values of each parameter were plotted at every 40 iterations (4hours) over 480 iterations (48h) and error bars were not displayed. An example of error bars representing standard deviation can be seen on S6 Fig. To help visualizing differences between each in silico conditions, min and max values of equivalent graphs across the various Figs were kept constant.

## Supporting information

**S1 Fig. Anteroposterior growth from the forelimb to the tail bud from 2 to 4 days of development.** A. Representative images of chicken embryos at stage HH13-, 18 and 23 corresponding to 48, 72 and 96 hours of incubation. The cephalic region of the embryo at stage HH18 removed prior to the picture. The green dotted line indicates the regions that was measured. B, plot of the mean length of the portion indicated in green in panel A. Dots represent the mean, error bars indicate standard deviation. Raw data of all plots provided in the S1 Spreadsheet.
(TIF)

**S2 Fig. Maintenance of neuroepithelial architecture requires actomyosin contractility.** A, Explants of the trunk are incubated in suspension with culture medium, culture medium with DMSO or culture medium with ROCK inhibitor (Y27632, 400µM) or Rac1 inhibitor (NSC27633, 400µM). B, Transversal sections with nuclear (DAPI, grey) and actin staining (Phalloidin, green). C, apicobasal length. D, number of pseudolayers of nuclei. E, straightness of the apical domain. F, shape of nuclei. G, position of mitoses, at scale with the tissue. H, position of mitoses, raw data (embryos/mitoses: $n_{control}$ = 2/42, $n_{DMSO}$ = 2/66, $n_{ROCK}$ = 3/153, $n_{RAC1}$ = 4/98; Kruskal-Wallis followed by multiple comparisons; ****, p<0.0001). Box and whiskers plot: the box extends from the 25th to the 75th percentile; the whiskers show the extent of the whole dataset. The median is plotted as a line inside the box. Raw data of all plots provided in the S1 Spreadsheet.
(TIF)

**S3 Fig. Inhibition of ROCK does not lead to cell death.** A-E, transversal sections of chick trunk explants after a 2-hour incubation in suspension in culture medium (A, control; n = 4 embryos), supplemented with DMSO (B-C, DMSO; n = 6 embryos) or with the ROCK inhibitor at 400µm (D-E; n = 5 embryos) followed by fixation, TUNEL assays revealed by NBT/BCIP (purple precipitate) and cryosectioning. C and E show portions of the extraembryonic tissues where positive cells for TUNEL were detected in the DMSO and ROCK inhibitor conditions. No TUNEL staining was found in the surface ectoderm, the neural tube, the paraxial mesoderm nor the notochord in either condition. D', is a zoom of the ROCK inhibitor condition where one can see that tissue deformation caused by the treatment is not accompanied by a massive induction of cell death (no TUNEL-positive cells). All images are at the same scale apart from the zoom in D' which is magnified twice compared to its cognate low magnification image shown in D.
(TIF)

**S4 Fig. Impairing cell-matrix adhesion leads to changes in nuclear, cell and tissue shape.** A, Explants of the trunk are incubated in suspension with culture medium (DMEM; n = 6 embryos) or culture medium with Dispase (0.2U/mL; n = 6 embryos). B, Transversal sections with nuclear staining (DAPI, grey) and immunostaining for Fibronectin, laminin or N-cadherin (green). C, apicobasal length. D, number of pseudolayers of nuclei. E, straightness of the apical domain. F, shape of nuclei. G, position of mitoses, at scale with the tissue. H, position of mitoses, raw data ($n_{DMEM}$ = 46, $n_{Dispase}$ = 46; **non-parametric t-test p = 0.0017). Box and whiskers plot: the box extends from the 25th to the 75th percentile; the whiskers show the extent of the whole dataset. The median is plotted as a line inside the box. Raw data of all plots provided in the S1 Spreadsheet.
(TIF)

**S5 Fig. Restricting lateral expansion does not compensate for the lack of apical contractility.** Simulations with passive apical springs, normal INM with different rates of exclusion of daughter cells as in Fig 5 and lateral walls. A, apicobasal length of the PSE (AB) and mean nuclear position along the AB axis (N) over time expressed in micrometers (see S5 Movie). B, Number of pseudolayers of nuclei along the AB axis. C, straightness of apical domain (net distance between the first and last apical point divided by the actual distance between these two points). D-F, net width of apical (magenta), nuclear (black) and basal (cyan) domains of the PSE with 50 (D), 40 (E) and 30% (F) of daughter cells being excluded from the 2D plane. For each domain, the net distance between the first and last point along the lateral axis is computed and its evolution plotted over time. Note that for all conditions the net width of each domain is constant over time confirming that the imposed lateral walls are indeed preventing lateral expansion. Raw data of all plots provided in the S1 Spreadsheet.
(TIF)

**S6 Fig. INM opposes apicobasal elongation, even with high random nuclear noise.** A, Mean apicobasal (AB, open circle) and nuclear-basal (N, closed circles) lengths with normal INM (black) and low INM (red). Note that mean tissue height goes from 75μm to 89μm in absence of INM. In the meantime, mean nuclear height goes from 51μm to 48μm representing a shift from being located within the most apical top third of the tissue with normal INM (51μm /75μm = 68%) to being located midway between the apical and basal domains (48μm/ 89μm = 54%). B-C, Net width of apical (magenta), nuclear (black) and basal (cyan) domains with normal INM (B) or low INM conditions (C). Note that low INM conditions lead to a shrinkage of the apical domain (arrow). Error bars represent standard deviation. Raw data of all plots provided in the S1 Spreadsheet.
(TIF)

**S1 Table. List of parameters used in each simulation presented on Figs 3–5, 7, S5 and S6.** Yellow, passive apical-apical springs; Green, contractile apical-apical springs; blue frame, fast update of basal points; magenta, high random nuclear movement (noise); light grey, no proliferation and no INM; dark grey, low INM; bold text, normal INM conditions. The following parameters were common to all simulations: radius of nucleus soft core (Rs = 5μm), radius of nucleus hard core (Rh_S = 1.5μm), radius of nucleus hard core during mitosis (Rh_M = 3.5μm), maximum distance between two consecutive apical points (a0 = 1/6*Rs), maximum distance between two consecutive basal points (b0 = 1/6*Rs). The relative strengths of the various forces was set as follows: stiffness of soft core of the nucleus (alpha_X = 1), stiffness of apical-nucleus spring (alpha_aX = 2), stiffness of nucleus-basal spring (alpha_bX = 2), stiffness of apical-apical spring (alpha_aS = 5), stiffness of apical-apical spring during G2 phase and

mitosis (alpha_aM = 10), magnitude of the apical-nucleus-basal alignment force (alpha_ab = 15).
(TIF)

**S1 Movie. (related to Fig 2).** Mosaic expression of membrane-GFP (green) and membrane-mCherry (red) into the chick neuroepithelium at stage HH14 at the level of the intermediate neural tube. Nuclei are counterstained with DAPI (grey).
(MOV)

**S2 Movie. (related to Figs 3 and 4).** All simulations start with 30 cells. Exclusion rate of daughter cells is set to 50% keeping the total cell number constant. Top panels: simulations with passive apical-apical springs without INM (left), with low INM (middle), with normal INM (right). Bottom panels: simulations with contractile apical-apical springs without INM (left), with low INM (middle), with normal INM (right). Note that in absence of INM there is a rapid shrinkage of the apical domain giving the tissue a pyramidal shape. Also, INM leads the rapid emergence of a low nuclear density region basally. Only hard cores of nuclei, apical points and basal points are represented. Soft cores of nuclei and springs are not displayed. Red, cells in mitosis; yellow cells in PRAM/active G2; black line, tracking of nuclei in PRAM/G2 and M. Each frame corresponds to one iteration of the simulation (circa. 6 minutes of biological time). Total duration 480 iterations (48h of biological time).
(MOV)

**S3 Movie. (related to Fig 5).** All simulations start with 30 cells with normal INM. From left to right, exclusion rate of daughter cells is set to 50% (constant cell number), 40%, 30%, 0% (all daughter cells added to the 2D plane). Top panels: simulations with passive apical-apical springs. Bottom panels: simulations with contractile apical-apical springs. Note that contractile apical springs mitigates buckling of the apical domain and feeds back into basal rearrangements. Red, cells in mitosis; yellow cells in PRAM/active G2; black line, tracking of nuclei in PRAM/G2 and M. Each frame corresponds to one iteration of the simulation (circa. 6 minutes of biological time). Total duration 480 iterations (48h of biological time).
(MOV)

**S4 Movie. (related to S5 Fig).** All simulations start with 30 cells with normal INM, passive apical springs and lateral walls (dashed lines). From left to right, exclusion rate of daughter cells is set to 50% (constant cell number), 40%, 30%.
(MOV)

**S5 Movie. (related to Fig 7).** All simulations start with 30 cells with normal INM and a 25-fold increase of random nuclear displacement at each iteration (noise) compared to previous simulations. From left to right, exclusion rate of daughter cells is set to 50% (constant cell number), 40% and 30%. The fourth condition at the far-right is with 30% of exclusion rate and fast update of basal points. Note that the global non-oriented force generated by increased noise is converted into apicobasal elongation. Also, allowing fast reorganization of the basal points promotes isotropic expansion of the apical and basal domain. Red, cells in mitosis; yellow, cells in PRAM/active G2; black line, tracking of nuclei in PRAM/G2 and M. Each frame corresponds to one iteration of the simulation (circa. 6 minutes of biological time). Total duration 480 iterations (48h of biological time).
(MOV)

**S6 Movie. (related to S6 Fig).** Simulations start with 30 cells. Exclusion rate is set to 50% (constant cell number). Left panel, normal INM. Right panel, low INM conditions. Note that without normal INM the apicobasal expansion is faster and that there is a shrinkage of the apical

domain.
(MOV)

**S1 Information. This file contains the details about the mathematical model.**
(PDF)

**S1 Spreadsheet. This file contains raw data for all plots in main and supplementary figures.**
(XLSX)

## Acknowledgments

We thank Drs Sara Merino-Aceituno, Samuel Tozer, Bertrand Benazeraf, Fabienne Pituello, Julie Batut, Ariane Trescases and Elisa Marti for critical reading of the manuscript. We are grateful to Dr. Diane Peurichard for helpful comments on the code and to Dr Angie Molina for providing the data to estimate the diffusion coefficient.

## Author Contributions

**Conceptualization:** Pierre Degond, Eric Theveneau.

**Data curation:** Evangeline Despin-Guitard, Fernando Duarte, Eric Theveneau.

**Formal analysis:** Marina A. Ferreira, Pierre Degond, Eric Theveneau.

**Funding acquisition:** Marina A. Ferreira, Pierre Degond, Eric Theveneau.

**Investigation:** Marina A. Ferreira, Evangeline Despin-Guitard, Fernando Duarte, Pierre Degond, Eric Theveneau.

**Methodology:** Marina A. Ferreira, Evangeline Despin-Guitard, Pierre Degond, Eric Theveneau.

**Project administration:** Pierre Degond, Eric Theveneau.

**Resources:** Pierre Degond, Eric Theveneau.

**Software:** Marina A. Ferreira, Pierre Degond.

**Supervision:** Pierre Degond, Eric Theveneau.

**Validation:** Marina A. Ferreira, Eric Theveneau.

**Visualization:** Marina A. Ferreira.

**Writing – original draft:** Marina A. Ferreira, Pierre Degond, Eric Theveneau.

**Writing – review & editing:** Marina A. Ferreira, Evangeline Despin-Guitard, Pierre Degond, Eric Theveneau.

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
