## [Decision Letter · Decision Letter 0]

3 Sep 2019

Dear Pierre,

Thank you very much for submitting your manuscript 'Interkinetic nuclear movements promote apical expansion in pseudostratified epithelia at the expense of apicobasal elongation' for review by PLOS Computational Biology. Your manuscript has been fully evaluated by the PLOS Computational Biology editorial team and in this case also by independent peer reviewers. The reviewers appreciated the attention to an important problem, but raised some substantial concerns about the manuscript as it currently stands. While your manuscript cannot be accepted in its present form, we are willing to consider a revised version in which the issues raised by the reviewers have been adequately addressed. We cannot, of course, promise publication at that time.

Best wishes.

Philip K Maini

Associate Editor

PLOS Computational Biology

Jason Haugh

Deputy Editor

PLOS Computational Biology

[LINK]

Reviewer's Responses to Questions

**Comments to the Authors:**

Reviewer #1: Studies of interkinetic nuclear migration are now en vogue in developmental biology, partly due to its association with cell differentiation and congenital defects. This timely manuscript by Ferreira and colleagues combines theoretical and experimental approaches to provide novel insight into this fundamental cell behaviour which could not have been achieved with either method alone. This manuscript is therefore expected to be of immediate interest to a broad field. A few limitations should be addressed:

1) Where are the Rock-inhibition methods detailed? The figure legend states explants were cultured in suspension in the presence of 400 µM Y27632. This is a very high concentration (manufacturer maximum solubility <200 mM in DMSO so would need >0.2% DMSO in culture) under sub-optimal culture conditions. It could cause cell death and delamination, which would explain changes observed in Supp Fig 2. Explant viability should be confirmed.

2) The phrase “apical constriction” appears to be used loosely to mean the average tension across the apical surface, rather than the actoymosin-dependent pulsatile contractility of individual cells as biologists will understand it. This needs to be clarified.

Rather than defining each cell’s apical surface as a single point, it may be more meaningful to describe it as the non-zero space between adjacent apical points (i.e. the springs, with the points representing cell-cell adhesions instead). Given the apical areas of cells change predictably during IKNM (which could be confirmed in the model used here), it may be meaningful to relate individual cell preferred apical lengths to IKNM phases. Alternatively, each cell could be related to an adjacent apical spring as its apical surface and contractility of individual springs could be varied.

3) The expansion and eversion of the apical surface in simulations under conditions of passive apical springs and low extrusion is very intriguing. It is reminiscent of the morphology of the neuroepithelium in the open neuropores of Rock-inhibited embryos (e.g. Escuin et al, J Cell Sci, 2015). However, it is difficult to picture the relevance of this eversion to the closed neural tube. The authors should establish whether imposing boundary conditions which limit dorso-ventral expansion, as is the case in vivo, compensates for the role of contractile apical springs (Figure 5).

4) The authors interpret the change in shape of the nuclei as indicating that “nuclei are slightly compressed along the dorsoventral axis, giving them an elongated form along the apicobasal axis”. The interpretation of this shape change is not unequivocal. For example, apical pulling of the nucleus during PRAM through an increasingly apically-crowded epithelium could favour a more elongated shape. Experimental verification of tension anisotropy (e.g. with tissue cut and deform experiments) should be provided. Can the authors comment on whether apical crowding alters PRAM rate in their simulation, for example due to increased drag?

5) Modelling a straight portion of PSE is very strategic, circumventing the emerging differences between IKNM in tissues with different geometries. In vivo most PSEs are curved. It would be informative to model or at least discuss the effect of basal curvature (e.g. as in the retina).

6) Cell adhesion is repeatedly mentioned throughout the manuscript. The model only incorporates the apical-most cell adhesions (i.e. tight junctions). In reality, other junctions such as N-cadherin adheres junctions extend laterally. Their likely impact on elongation and the requirement for active apical tension should be discussed.

Minor comments:

7) The authors may wish to cite the observation in Xenopus embryos that their neuroepithelium elongates apicobasally during closure despite not being pseudostratified (e.g. see Inoue et al, Biomech Model Mechanobiol,2016). This supports their conclusion that the two processes are likely to be independent.

8) CAKUT should be defined where it is first used in the introduction, not in the discussion.

9) The representative images shown in Supp Fig 2 suggest the dorsoventral length (i.e. roof plate to floor plate) of Rock-inhibited embryos is longer than controls. Is this reproducible? What might this mean (given 2 hours is likely to be too short for it to be the consequence of changes in IKNM)?

10) The authors discuss various potential explanations for why apical crowding only occurs after the neural tube closes. The apical surface of the open posterior neuropore is longer than the length of the closed lumen. Again, the change in boundary conditions between the open and closed neural tube may affect apical crowding, stratification and elongation.

Reviewer #2: The manuscript by Ferreira and colleagues examines the potential roles of interkinetic movements of nuclei (INM) for the growth and shape of pseudostratified epithelia using the developing chick neural tube as a model. They use 2D computational modelling and simulations together with embryo observations and some limited experimental manipulation. As the neural tube grows, they observe changes in nuclear shape, and cell size and shape and number of nuclei and of pseudostratified layers. The model examines how these localized changes could result in tissue level changes. Overall they propose that INM is important for the expansion of the apical domain of the epithelium. Furthermore, their data suggests that the apicobasal elongation of neuroepithelial cells is not an emergent property but instead requires a separate elongation program.

The authors explain in detail the assumptions made for the mathematical modelling and the parameters used for the simulations. They make some insightful observations which may be of more general importance and may also be relevant to other pseudostratified epithelia. The data are well-presented and discussed.

The experimental manipulations are limited to exposure of neuroepithelium to ROCK inhibitor and the authors may consider using inhibitors of additional cellular processes.

Reviewer #3: In this paper, an original 2D off-lattice agent-based model for the dynamics of pseudostratified epithelia is presented.

Computational simulations of the model are combined with experimental results on the growth of the chick neuroepithelium to assess the impact of interkinetic movement versus other cytoskeleton-dependent processes, such as adhesion and mitosis, on the dynamics of pseudostratified epithelia.

In the mathematical model, each cell is approximated as a nucleus, an apical point and a basal point. The cell nucleus is represented as an inner hard sphere surrounded by an outer soft sphere. The deformation occurring when cell nuclei are pressed against each other is incorporated into the model by allowing the soft spheres to overlap, while the consequent repulsion between the nuclei is modelled by imposing a dynamical non-overlapping constraint between the hard spheres. The cell cytoplasm is seen as a viscoelastic material modelled through as a set of springs linking the nuclei to the apical and basal points.

Cell division and both cell-cell and cell-matrix interactions are incorporated into the agent-based model through a set of mechanical and behavioural rules, which are elegantly integrated with an energy minimisation algorithm ensuring that the non-overlapping constraint between the hard parts of the cell nuclei is satisfied.

The paper is pleasant to read, the presentation is clear and the results are interesting. The model appears to be original, based on sound modelling strategies and of potential interest to applied mathematicians and physicists working on the mathematical modelling of tissue mechanics. The short summary of the model presented in the paper is effective and nicely written. Moreover, a clear and detailed description of the model is provided in the Supplementary Information along with a precise account of the model parameterisation and the details of computational simulations. As a mathematician, I cannot comment on the quality of the experimental results but the conclusions drawn from the outcomes of the model appear to be coherent and interesting.

**Have all data underlying the figures and results presented in the manuscript been provided?**

Reviewer #1: No: Data that underlies graphs not provided in spreadsheet form as supporting information

Reviewer #2: Yes

Reviewer #3: Yes

PLOS authors have the option to publish the peer review history of their article (what does this mean?). If published, this will include your full peer review and any attached files.

Reviewer #1: Yes: Gabriel L Galea

Reviewer #2: No

Reviewer #3: No

---

## [Decision Letter · Decision Letter 1]

17 Nov 2019

Dear Dr Degond,

We are pleased to inform you that your manuscript 'Interkinetic nuclear movements promote apical expansion in pseudostratified epithelia at the expense of apicobasal elongation' has been provisionally accepted for publication in PLOS Computational Biology.

In the meantime, please log into Editorial Manager at https://www.editorialmanager.com/pcompbiol/, click the "Update My Information" link at the top of the page, and update your user information to ensure an efficient production and billing process.

One of the goals of PLOS is to make science accessible to educators and the public. PLOS staff issue occasional press releases and make early versions of PLOS Computational Biology articles available to science writers and journalists. PLOS staff also collaborate with Communication and Public Information Offices and would be happy to work with the relevant people at your institution or funding agency. If your institution or funding agency is interested in promoting your findings, please ask them to coordinate their releases with PLOS (contact ploscompbiol@plos.org).

Thank you again for supporting Open Access publishing. We look forward to publishing your paper in PLOS Computational Biology.

Sincerely,

Philip K Maini

Associate Editor

PLOS Computational Biology

Jason Haugh

Deputy Editor

PLOS Computational Biology

Reviewer's Responses to Questions

**Comments to the Authors:**

Reviewer #1: The authors have meaningfully amended the text and provided new experimental as well as in silico data which are well placed and address all my comments.

I confirm the authors interpreted my question regarding PRAM "rate" as intended and the simulations described in the responses to reviewers address it. This data could be useful in simulations of the timing of neurogenesis (e.g. Hadjivasiliou et al, Dev Cell, 2019) so I hope it will be made available, for example by publishing the peer review history.

Reviewer #2: This is a very nice manuscript and the authors have responded in detail to the comments made.

**Have all data underlying the figures and results presented in the manuscript been provided?**

Reviewer #1: Yes

Reviewer #2: Yes

PLOS authors have the option to publish the peer review history of their article (what does this mean?). If published, this will include your full peer review and any attached files.

Reviewer #1: Yes: Gabriel L Galea

Reviewer #2: No

---

## [Editor Report · Acceptance letter]

17 Dec 2019

PCOMPBIOL-D-19-00915R1 

Interkinetic nuclear movements promote apical expansion in pseudostratified epithelia at the expense of apicobasal elongation

Dear Dr Degond,

I am pleased to inform you that your manuscript has been formally accepted for publication in PLOS Computational Biology. Your manuscript is now with our production department and you will be notified of the publication date in due course.

With kind regards,

Bailey Hanna
